# 10-Year Clinical, Functional, and X-ray Follow-Up Evaluation of a Novel Posterior Percutaneous Screw-Rod Instrumentation Technique for Single-Level Pyogenic Spondylodiscitis

**DOI:** 10.3390/tropicalmed6030159

**Published:** 2021-09-01

**Authors:** Enrico Pola, Luigi Aurelio Nasto, Valerio Cipolloni, Debora Colangelo, Antonio Leone, Alfredo Schiavone Panni

**Affiliations:** 1Department of Orthopaedics, A.O.U. “Vanvitelli” University Hospital, “Luigi Vanvitelli” University, Via del Sole 10, 80138 Naples, Italy; valeriocipolloni@gmail.com (V.C.); debora.colangelo@gmail.com (D.C.); alfredo.schiavonepanni@unicampania.it (A.S.P.); 2Department of Paediatric Orthopaedics, IRCCS Istituto “G. Gaslini”, Via G. Gaslini 5, 16147 Genova, Italy; luigiaurelionasto@gaslini.org; 3Department of Medical Imaging, Radiotherapy and Haematology, Università Cattolica del Sacro Cuore di Roma, “A. Gemelli” University Hospital, l.go A. Gemelli 1, 00168 Roma, Italy; antonio.leone@policlinicogemelli.it

**Keywords:** pyogenic spondylodiscitis, spinal infections, surgical treatment of pyogenic spondylodiscitis, minimally-invasive spinal surgery, percutaneous posterior spinal stabilization

## Abstract

Medical treatment with antibiotic therapy remains the mainstay of treatment for pyogenic spondylodiscitis (PS). Nevertheless, orthopaedic treatment is also very important in relieving pain, preventing neurological damage, and avoiding development of spinal deformities (e.g., scoliosis, kyphosis) due to spinal instability. Rigid thoracolumbosacral orthosis (TLSO) bracing is often needed in patients with PS, and average duration of treatment of 3 to 4 months. However, TLSO bracing can be poorly tolerated and limit ability of the patient to go back to a normal life. In 2004 our group developed an alternative surgical treatment to TLSO bracing by percutaneous posterior screw-rod bridge instrumentation of the infected level. This treatment allows early and free mobilization of the patients and is associated with faster recovery, lower pain scores and improved quality of life as previously reported. Herein, we report the clinical outcome of the first 3 patients who have completed the 10 years follow-up mark after the procedure. A case report is also described and details of the procedure are provided.

## 1. Introduction

Spinal infections represent 3–5% of all cases of osteomyelitis, with incidence ranging from 0.5 to 5.8/100,000 inhabitants/year [1,2]. The term pyogenic spondylodiscitis (PS) refers to an infection process of a disc space and adjacent vertebral bodies caused by pyogenic bacteria (e.g., *S. aureus*, coagulase-negative staphylococci). If left untreated, PS can have serious consequences including worsening pain, onset of neurological deficit, development of spinal deformities (i.e., kyphosis, scoliosis) due to disruption of structural integrity of the spine, local or haematogenous spread of the infection, multiple organ failure, and death [3,4]. Risk of complications and poor outcome is higher in elderly or chronically debilitated patients (e.g., immunocompromised patients, ev drug abusers, or patients with metastatic cancer). Four to six weeks of ev antibiotic therapy followed by a variable oral course of antibiotics is the mainstay of treatment [3,5]. Microbiologic diagnosis can be obtained through blood cultures, CT-guided, percutaneous or open biopsy and it is of the utmost importance in directing correct antibiotic treatment.

Orthopaedic management of PS is aimed at relieving pain, treating any actual or impending neurological damage, and preventing development of spinal deformity or neurological deficit due to compression on spinal cord/nerves [6]. Rigid bracing with a thoracolumbosacral orthosis (TLSO) is generally prescribed for thoracic or lumbar spine infections. TLSO bracing can be very helpful in relieving pain and preventing development of spinal deformities; however, intensive (24/7) and long-term (3 to 4 months) treatment is often needed to achieve full healing [7,8]. In 2014, we published our early results with a novel percutaneous posterior screw-rod fixation technique as alternative approach to TLSO bracing. Our data showed that surgical stabilization was associated with faster recovery, lower pain scores, and improved quality of life over TLSO rigid bracing at 1 and 3 months after treatment. No neurological complication nor spinal deformity related to surgery was noted, although the longest available follow-up at that time was 9 months [9]. Over the years we have continued to offer this alternative treatment to our patients with results similar to those of our previously published work.

The aim of this case report and brief communication is to provide clinical, functional, and X-ray follow-up data of the first 3 patients who have completed the 10-years follow-up mark. We believe these data can be valuable to clinicians and orthopaedic surgeons alike in informing their treatment choices for PS.

## 2. Materials and Methods

Following Institutional Review Board (IRB) approval, we contacted all consecutive patients who underwent posterior percutaneous screw-rod instrumentation for PS at our institution and completed the 10-year follow-up mark. Patients were retrieved from a longitudinal, singe-center, prospective database of spinal infections that was started at our institution in January 2008. All patients gave their written informed consent at the enrolment into the database.

The diagnosis of PS was made on clinical (i.e., symptoms suggestive of PS with laboratory abnormalities of increased white blood cell count, erythrocyte sedimentation rate, and C-reactive protein), radiological (i.e., abnormal MRI and CT scans), and microbiological (i.e., isolation of the causative agent and/or typical histologic pattern) grounds. Indications for surgical treatment through percutaneous posterior screw-rod instrumentation were as follows: single-level (one-disc) lower thoracic (T9-T12) or lumbar PS, known infectious agent, no actual or impending neurologic compromise, no systemic compromise (absence of relevant changes in vital parameters), and no gross spinal instability. Mechanical instability was defined as less than 25% change of segmental kyphosis at the infection level in standing and supine lateral view X-rays. Patients were given the option to choose between TLSO bracing or surgery, therefore no randomization process was used for treatment allocation.

Surgical operation consisted of a percutaneous posterior instrumentation with a four to eight screw construct bridging the level of the infection. Screws were placed in the vertebral bodies adjacent to the infected level and rods were passed percutaneously after appropriate contouring. Patients were allowed free mobilization as pain tolerated immediately after surgery. Antibiotic therapy was prescribed as per consultation with infectious diseases specialists at our institution. During treatment C-reactive protein (CRP), erythrocyte sedimentation rate (ESR), and complete blood count were obtained at the time of diagnosis and at 4, 8, 12, and 24 weeks follow-up. Lateral spine standing X-rays were obtained before surgery, before hospital discharge and at each follow-up visit. Patients were also asked to fill in Visual Analog Scale (VAS), Short-Form 12 (SF-12), and EuroQol five dimension (EQ-5D) questionnaires at each follow-up visit. A healed infection was considered after a complete course of antibiotic therapy (minimum 6 weeks), negative blood inflammatory markers, absent clinical findings (i.e., pain, paraspinal muscle tenderness, fever) 12 months after the end of treatment [10].

For the current analysis (10 years follow-up), clinical and functional outcome was assessed through VAS, SF-12, and EQ-5D questionnaires. Whole spine standing antero-posterior and lateral X-rays of the spine were obtained along with CT scan of the previously treated area of the spine.

### Statistical Analysis

Data are expressed as mean with ranges unless stated otherwise; counts and percentages are used as appropriate. Data were analyzed using Microsoft Office Excel 2016 Professional (Microsoft, Redmond, WA, USA).

## 3. Results

A total of 214 patients with confirmed pyogenic spondylodiscitis (PS) were included in our institutional database from January 2008 through December 2018. Among these, 32 patients received surgical treatment through percutaneous screw-rod instrumentation, 4 of them more than 10 years ago. All four patients were contacted for the current analysis. One patient suffering from severe heart failure died 2 years before the current study and was not included in the study. Demographics and baseline data of the three patients included in the current study are summarized in Table 1.

The level of the infection was T11-T12 in two patients and T8-T9 in one patient. *S. aureus MSSA* was the causative agent in on patient, *S. aureus MRSA + E. coli* was the causative agent in the second patient, while *S. hominis* was isolated in the last patient. Antibiotic therapy is listed in Table 2.

All three patients underwent percutaneous posterior screw-rod instrumentation. Mean operative time was 75 (55–115) minutes. There were no intraoperative complications; intraoperative blood loss was minimal (<50 mL). Complete healing was achieved in all patients; average healing time was 68.3 (55–75) days. Pre- and post-treatment VAS, SF-12 mental and physical component, and EQ-5D score changes are shown in Figure 1, Figure 2, and Figure 3, respectively. Significant improvement in all scores was observed during the first months after the procedure and antibiotic therapy. Improvements were maintained at the 10 years follow-up.

Average pre-operative segmental kyphosis was 20.2° (15.9–22.4). Immediately after surgery segmental kyphosis was 17.0° (13.6–21.4), which was maintained at 15° (11.3–20.4) at 10 years follow-up (Figure 4). There was no evidence of screw loosening and complete fusion at the infected level was noted in all three patients.

### Case Report

In June 2009, a 68-year old man presented to you hospital with a 1 month history of worsening axial back pain at the thoracolumbar junction. Past medical history of the patient was positive for high blood pressure and benign prostatic hyperplasia. At first the patient was assessed by his general practictioner and was prescribed oral antibiotic therapy with ciprofloxacin for a suspect urinary tract infection. Therapy was unsuccessful and patient developed fever, night sweats and fatigue over the course of the following 15 days. The patient was admitted to our hospital with the suspect diagnosis of pyogenic spondylodiscitis. Spinal X-ray, CT and MRI scans were obtained (Figure 5A,B). Imaging showed destructive lesion at T11-T12 with disc and endplate involvement, suggestive of pyogenic spondylodiscitis type B3.1 [11]. A CT guided biopsy of the lesion was positive for *S. hominis* pyogenic infection. The patient was started on intravenous teicoplanin and ceftriaxone therapy which was continued for 4 weeks and followed by other 4 weeks of iv therapy with teicoplanin and co-trimoxazole. At the end of iv antibiotic therapy the patient was prescribed oral therapy with co-trimoxazole and rifampicin for 4 more weeks. Before starting antibiotic therapy the patient was prescribed a rigid TLSO brace to relieve pain and prevent development of spinal deformity. The brace was poorly tolerated by the patient and for this reason operative treatment with percutaneous posterior screw-rod instrumentation was offered. Posterior fixation from T10 to L1 was performed uneventfully (Figure 5C,E). The patient was allowed free mobilization soon after surgery. Clinical and laboratory parameters improved significantly over the course of the follow-up and complete healing of the infection was determined at 3 months after the diagnosis. The patient is currently 80-years old and completed the 10-years follow-up after the procedure. He lives indepedently at home, does not complain of any back pain. The imaging obtained at 10 years follow-up shows complete healing of the lesion with fusion of the vertebral bodies. No mobilization of the instrumentation is noted (Figure 5D,F).

## 4. Discussion

This is the first study to report long-term outcomes of percutaneous posterior screw-rod instrumentation for thoracolumbar pyogenic spondylodiscitis. Our results confirm some of the observations reported in our previously published study [9]. More specifically, complete healing of the infection was noted in all of our 10 years follow-up patients. No superinfection of the instrumentation was noted. Patient reported outcome measures remained stable over the course of the 10 years follow-up; similarly, there was no increase in segmental kyphosis, thus showing that no spinal deformity developed over the years.

Antibiotic therapy remains the mainstay of treatment for pyogenic spondylodiscitis [4]. Nevertheless, orthopaedic treatment is also important in order to relieve pain, prevent development of neurological damage and spinal deformity. From this perspective prescription of a TLSO brace or any other sort of spinal immobilization is often needed [3]. While treatment with TLSO brace remains safe and effective, it can be cumbersome and poorly tolerated by patients. This becomes even more important in patients with multiple comorbidities or other medical issues. Furthermore, duration of treatment with TLSO has not been determined in literature and mostly depends on clinical judgment of the treating surgeon [7,8]. Risk of worsening of deformity is higher in the first 6 to 8 weeks after diagnosis. Therefore, average duration of treatment with TLSO is 3 to 4 months.

In 2009 we started offering our patients an alternative treatment to TLSO bracing. Our treatment consisted of percutaneous posterior screw-rod bridge instrumentation of the infected level [9]. The surgical instrumentation works as an internal brace and avoids usage of any external immobilization device. The patient is allowed early and free mobilization and can go back to his daily activities as soon as tolerated. Over the years we have operated a total of 32 patients with encouraging results. In our first report published in 2014, patients undergoing posterior percutaneous stabilization were compared to patients treated with TLSO brace at short term (i.e., 9 months) follow-up. Complete infection healing with antibiotic therapy was achieved in all patients with no significant differences in healing time between the two groups. However, surgically treated patients had significantly lower VAS scores at 1 and 3 months and statistically significant faster improvements of SF-15 and EQ-5D scores at 1 and 3 months compared to TLSO bracing treated patients. At the time of the last follow-up reported in the study (i.e., 9 months) both groups (i.e., surgery and TLSO bracing) showed similar VAS, SF-12 and EQ-5D scores [9]. In this manuscript we report the results of the first 3 patients who have completed the 10-years follow-up mark. Although it is not common to find published results for surgical procedures at long follow-up (i.e., >5–10 years), this is a very important aspect of any surgical procedure. Our data show that improvements achieved with healing of the infection and surgery are maintained at 10 years follow-up. No screw loosening or superinfection was observed in our patients. Kyphosis and mechanical stability of the spine was maintained and patients showed no significant worsening pain or spine disability over time.

Interest in minimally invasive procedures has increased in recent years. Even in the field of spinal infection, several minimally invasive procedures have been developed and the field is rapidly expanding [12,13,14,15,16,17]. The range of interventions proposed varies from percutaneous injection of local antibiotics for treatment of recalcitrant infections [17], to percutaneous suction and irrigation of the infected disc space [12,13,16], from minimally-invasive surgical techniques for infection debridement [14,15], to percutaneous posterior bridge stabilization of the infected level [9]. The advantages of minimally invasive techniques are related to the shorter length of stay, decrease perioperative morbidity and quicker recovery. However, while several techniques have been proposed, there is still a paucity of data on long term outcomes of these techniques [18]. To the best of our knowledge, this is the first study to report long term outcome of a minimally invasive technique used in patients with pyogenic spondylodiscitis.

Our study has some limitations. We acknowledge the fact that the number of patients presented in this report is limited and longer follow-up of other patients will be needed in order to consolidate our results. Furthermore, follow-up clinical outcome of patients treated with TLSO brace was not available at the time of this report. Nevertheless, in a previous publication on this technique a formal comparison with a conservative treatment group has been published, although with a shorter follow-up, demonstrating quicker recovery of the surgically treated group [9].

## 5. Conclusions

In conclusion this brief report confirms and expands on our previously published results on a percutaneous posterior fixation technique for pyogenic spondylodiscitis. Our technique allows early mobilization of the patients, decreases pain and prevents development of long-term spinal deformities due to infection-driven spinal instability. No significant complications were noted in our first three patients who have completed the 10-year follow-up and overall satisfaction rate of the patients was high.

## Figures and Tables

**Figure 1 tropicalmed-06-00159-f001:**
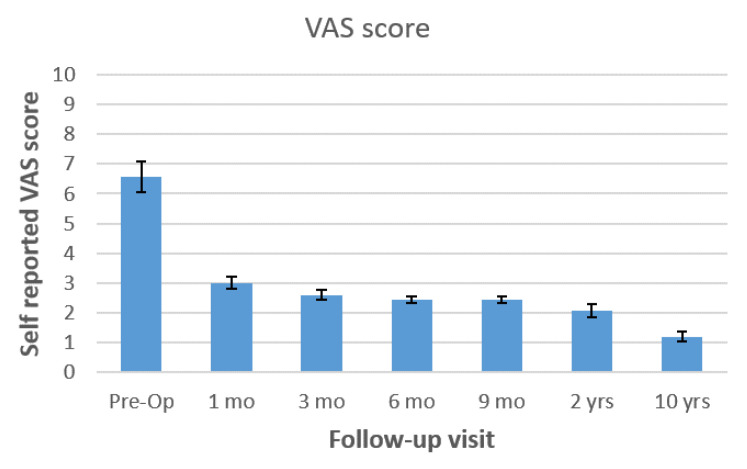
Pre- and post-treatment self-reported VAS scores for back pain (*n* = 3, 10 years follow-up) are shown at 1, 3, 6, 9, months, 2 years and 10 years after surgical treatment with percutaneous posterior fixation.

**Figure 2 tropicalmed-06-00159-f002:**
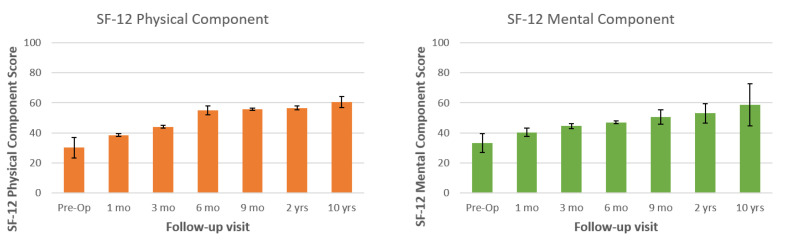
Pre- and post-treatment SF-12 physical and mental component scores (*n* = 3, 10 years follow-up) are shown at 1, 3, 6, 9, months, 2 years and 10 years after surgical treatment with percutaneous posterior fixation.

**Figure 3 tropicalmed-06-00159-f003:**
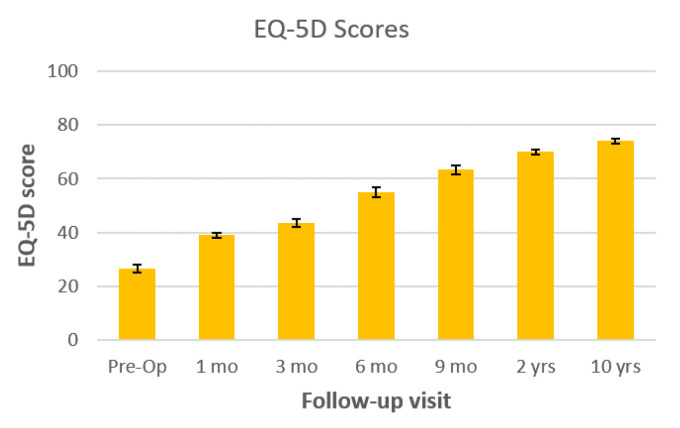
Pre- and post-treatment EQ-5D scores (*n* = 3, 10 years follow-up) are shown at 1, 3, 6, 9, months, 2 years and 10 years after surgical treatment with percutaneous posterior fixation.

**Figure 4 tropicalmed-06-00159-f004:**
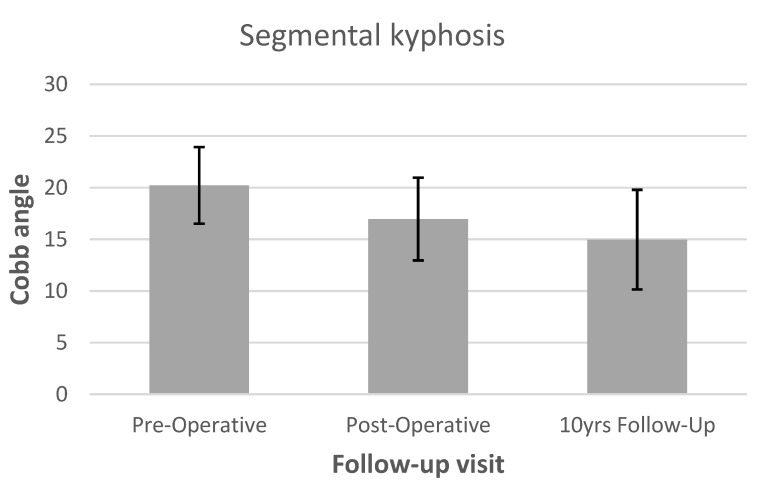
Pre-operative, immediate post-operative and 10 years follow-up segmental kyphosis level.

**Figure 5 tropicalmed-06-00159-f005:**
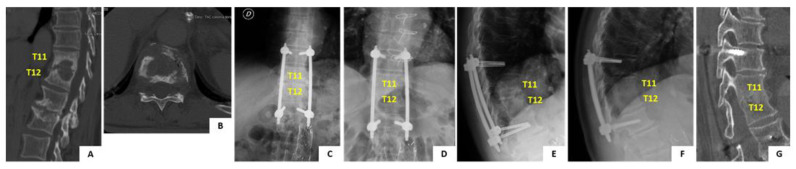
Pre- and post-operative imaging of the case report detailed in the manuscript. Panels (**A**,**B**) show sagittal and axial CT scan imaging of the T11–T12 infected level. Panels (**C**,**D**) show comparison of standing AP X-ray imaging of the spine immediately after surgery and at 10 years follow-up. Instrumentation is noted with pedicle screws and rods. The instrumentation is bridging the infected level. Panels (**E**–**G**) show comparison of standing lateral view X-ray imaging of the spine immediately after surgery and at 10 years follow-up. No mobilization of the instrumentarion is noted, the infected level is completely fused and segmental kyphosis is preserved.

**Table 1 tropicalmed-06-00159-t001:** Baseline demographics.

	At Diagnosis (*n* = 3)
Age at diagnosis (yrs)	69 (68–70)
Sex (M:F)	2:01
Diagnostic delay (days)	48.7 (21–80)
Presenting symptoms	
Axial back pain (n)	3
Radiating pain (UL–LL) (n)	1
Fever (n)	3
Weight loss (n)	1
Neurological impairment (n)	0
WBC count (/mm^3^)	52.1 (32.6–78.4)
C-reactive protein (mg/dL)	63.7 (56.8–67.6)
Comorbidities	
Cigarette smoking (n)	2
IV drug abuse (n)	0
Long term steroid therapy (n)	2
Alcohol consumption (n)	0
Immunodepression (n)	2
Diabetes (n)	1
Cancer (n)	0
CVD (n)	0
Liver cirrhosis (n)	0
Chronic renal failure (n)	0
Other (n)	3
Comorbidities	3.3 (2–4)

Data are expressed as means (range) unless stated otherwise. M, males; F, females; UL, upper limbs; LL, lower limbs; WBC, white blood cells; IV, intravenous; CVD, cardiovascular disease.

**Table 2 tropicalmed-06-00159-t002:** Spinal level, causative agent, antibiotic therapy.

	Infection Level	Causative Agent	Antibiotic Therapy (Duration, Route)
Case #1	T11-T12	*S. aureus (MSSA)*	Oxacillin + Rifampicin (2 weeks, iv) Levofloxacin + Rifampicin (10 weeks, po)
Case #2	T8-T9	*S. aureus (MRSA)*, *E. coli*	Meropenem (2 weeks, iv) Teicoplanin + Rifampicin + Levofloxacin (8 weeks, iv) Levofloxacin (21 weeks, po)
Case #3	T11-T12	*S. hominis*	Teicoplanin + Ceftriaxone (4 weeks, iv) Teicoplanin + Co-trimoxazole (4 weeks, iv) Co-trimoxazole + Rifampicin (4 weeks, po)

## Data Availability

Data available on request due to privacy restrictions.

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
