# Peer review of "10-Year Clinical, Functional, and X-ray Follow-Up Evaluation of a Novel Posterior Percutaneous Screw-Rod Instrumentation Technique for Single-Level Pyogenic Spondylodiscitis"

_tropicalmed, 2021, doi:10.3390/tropicalmed6030159_

Round 1
Reviewer 1 Report
#1. There is a description of the postoperative angle but no explanation if the screw loosened and the bone union or did not.
#2. Figure 5.
It is difficult to understand the position of the level and the screw, and the scale is different.
#3.Table 3
 The information about the duration of antibiotic use and whether switched to oral is very important for clinicians. Let describe in detail.
Author Response
Dear Reviewer,
Thank you for taking the time to review our manuscript and for your insightful comments. We appreciate the opportunity to improve the quality of our work through your comments. We hope we have made the necessary changes, however we will be happy to make further changes if felt necessary.
Please, find below our comments to your suggestions (italics).
#1. There is a description of the postoperative angle but no explanation if the screw loosened and the bone union or did not.
We did not observe any screw loosening in the three patients reported in this brief report. Furthermore, CT scan performed at 10 years demonstrated complete fusion of the infected level and no mobilization of the screws. We have added a sentence to the manuscript to clarify this. CT scan of one patient is also shown in Figure 5, panel G, and it shows complete fusion of the infected level. Figure 5, panel F, also shows a standing X-ray at 10 years with no evidence of screw loosening.
#2. Figure 5. It is difficult to understand the position of the level and the screw, and the scale is different.
Thank you for pointing this out. We have revised the figure by rescaling all pictures and also adding reference to the spinal levels. The figure is small in size because of page size constraints. However, the original file we have is larger and higher quality. We will be happy to provide the file to the editorial agency for final publication.
#3.Table 3 The information about the duration of antibiotic use and whether switched to oral is very important for clinicians. Let describe in detail.
Thank you for your comment. We have added information about duration and administration route in the Table. Antibiotic therapy was prescribed as per our infectious disease specialists advice.
Reviewer 2 Report
The authors report the clinical outcome of the first 3 patients who have completed the 10 years follow-up mark after the procedure. A case report is also described and details of the procedure are provided.
- Even though the follow-up time is long enough, the samples is really small to conclude the meaningful outcomes. I would suggest the authors present the paper as the case report only without statistics analysis due to the inconvincible results.
- The authors should declare the setting of indication for the surgery. Classical references are recommended to support the description.
- A p value should be highlight in the corresponding figures in order to significance of the results.
- Can the authors have a further comparison between the patients in short term follow up and long term follow up including the surgical outcomes, functional improvement? I think the differences may provide us novel idea.
Author Response
Dear Reviewer,
Thank you for taking the time to revise our manuscript and for your insightful comments. We appreciate the opportunity to improve the quality of our work through you comments. We hope we have made the necessary changes, however, we will be happy to make further changes if felt necessary.
Please, find below our comments to your suggestions (italics).
The authors report the clinical outcome of the first 3 patients who have completed the 10 years follow-up mark after the procedure. A case report is also described and details of the procedure are provided.
We appreciate your positive comments about our work.
Even though the follow-up time is long enough, the samples is really small to conclude the meaningful outcomes. I would suggest the authors present the paper as the case report only without statistics analysis due to the inconvincible results.
We completely agree with your comment and we acknowledge the fact that with an n = 3 no meaningful statistics can be performed. For this reason no p value is reported in the manuscript. Our manuscript is essentially a case report enriched with the data of 3 other patients who have reached the 10 year follow-up mark. We have made sure this is clearly stated in the aim of the manuscript (lines 59-62). The statistical analysis section (lines 104-108) only states how means and ranges were computed but there is no reference at any sort of statistical analysis which, we agree, would be meaningless with only 3 patients. The manuscript has been classified as "case report / brief report" in line 1, page 1, to the benefits of the readers.
The authors should declare the setting of indication for the surgery. Classical references are recommended to support the description.
Indications for surgery were: single-level (one disc) lower thoracic (T9-T12) or lumbar pyogenic spondylodiscitis, known infectious agent, no actual or impending neurologic compromise, no systemic compromise (absence of relevant changes in vital parameters), and no gross spinal instability (defined as more than 25% change in segmental kyphosis at the infection level in standing and supine x-ray). With these parameters, patients we given the option to choose between rigid TLSO brace treatment of surgery with percutaneous posterior stabilization. We have added details of the indications for surgery in lines 75-82 of the manuscript.
A p value should be highlight in the corresponding figures in order to significance of the results.
We appreciate your comment. We decided not to add p values to the figures because of the small number of patients involved in this case report / brief report. We felt that performing an ANOVA test on three patients would not be helpful or provide any meaningul information. While a formal statistical analysis of this procedure at short term has been published (Nasto LA, et al. Spine J, 2014), clearly more patients will be needed for a definitive undestanding of the benefits of this procedure in the long term. Data collection is underway and we will definitively report in the near future data of our patients once a significant number of them will reach the 10-years follow-up mark. Meanwhile we thought that showing the graph of VAS score, SF-12 and EQ-5D scores of the first 3 patients could at least provide a glimpse of what the trend is to the general readeship of the Journal.
Can the authors have a further comparison between the patients in short term follow up and long term follow up including the surgical outcomes, functional improvement? I think the differences may provide us novel idea.
Thank you for your comment. Short term comparison between surgery and conservative treatment with TLSO brace was part of a previous publication from our group (Nasto LA, et al, Spine J 2014). In our first report published in 2014, patients undergoing posterior percutaneous stabilization were compared to patients treated with TLSO brace at short term (i.e. 9 months) follow-up. Complete infection healing with antibiotic therapy was achieved in all patients with no significant differences in healing time between the two groups. However, surgically treated patients had significantly lower VAS scores at 1 and 3 months and statistically significant faster improvements of SF-15 and EQ-5D scores at 1 and 3 months compared to TLSO bracing treated patients. At the time of the last follow-up reported in the study (i.e. 9 months) both groups (i.e. surgery and TLSO bracing) showed similar VAS, SF-12 and EQ-5D scores. In summary, our study showed that surgery can offer a quicker recovery and better quality of life for patients in the first 1-3 months after diagnosis. At longer follow-up, once infection is healed and TLSO brace has been removed results between surgery and conservative treatment are similar. We have expanded our discussion and included a short paragraph detailing our previous results on this surgical technique and further discussing differences with conservative treatment.
Reviewer 3 Report
Dear Authors
I have reviewed your paper with great interest.
I will accept your paper after a minimal revision.
My revision is:
Title: Very Good
Abstract: Very Good
Introduction and AIM: The problem and the aim are well descripting.
Marterials, Patients and methods and statistics: All good.
Results: Focus on and well described.
Discussion and Thread: effectiveness Focus ON.
Cite and discuss this techinique to reduce the infection before the short segmental fixation by peduncular screws and rods:
Bonura EM, Morales DJO, Fenga D, Rollo G, Meccariello L, Leonetti D, Traina F, Centofanti F, Rosa MA. Conservative Treatment of Spondylodiscitis: Possible Therapeutic Solution in Case of Failure of Standard Therapy. Med Arch. 2019 Feb;73(1):39-43. doi: 10.5455/medarh.2019.73.39-43. PMID: 31097859; PMCID: PMC6445632
References: Well chosen but to improve
Figures and Table: Very Good.
Author Response
Dear Reviewer,
Thank you for taking the time to revise our manuscript and for your insightful comments. We appreciate the opportunity to improve the quality of our work through your insightful comments. We hope we have made the necessary changes, however we will be happy to make further changes if felt necessary.
Please, find below our comments to your suggestions (italics).
Cite and discuss this techinique to reduce the infection before the short segmental fixation by peduncular screws and rods:
Bonura EM, Morales DJO, Fenga D, Rollo G, Meccariello L, Leonetti D, Traina F, Centofanti F, Rosa MA. Conservative Treatment of Spondylodiscitis: Possible Therapeutic Solution in Case of Failure of Standard Therapy. Med Arch. 2019 Feb;73(1):39-43. doi: 10.5455/medarh.2019.73.39-43. PMID: 31097859; PMCID: PMC6445632
References: Well chosen but to improve
Thank you for your comments and for pointing at this very interesting technique and reference. We have added the suggested reference to our reference list and also improved the discussion section of the manuscript by highlighting the different other minimally invasive techniques available for treatment of pyogenic spondylodiscitis.
Round 2
Reviewer 2 Report
The comments are carefully responded and the paper is significantly improved.